# Ultra-Thin and Lithography-Free Transmissive Color Filter Based on Doped Indium Gallium Zinc Oxide with High Performance

**DOI:** 10.3390/mi13081228

**Published:** 2022-07-31

**Authors:** Xiangrui Fan, Shengyao Wang, Dongdong Xu, Gaige Zheng

**Affiliations:** 1School of Electronics & Information Engineering, Nanjing University of Information Science & Technology, Nanjing 210044, China; jsmetamaterials@163.com; 2Jiangsu Collaborative Innovation Center on Atmospheric Environment and Equipment Technology (CICAEET), Nanjing University of Information Science & Technology, Nanjing 210044, China; syaowang2022@163.com (S.W.); ddongxu2022@163.com (D.X.); 3Jiangsu Key Laboratory for Optoelectronic Detection of Atmosphere and Ocean, School of Physics and Optoelectronic Engineering, Nanjing University of Information Science & Technology, Nanjing 210044, China

**Keywords:** color filter, indium gallium zinc oxide, lithography-free

## Abstract

A kind of ultra-thin transmissive color filter based on a metal-semiconductor-metal (MSM) structure is proposed. The displayed color can cover the entire visible range and switches after H_2_ treatment. An indium gallium zinc oxide (IGZO) semiconductor was employed, as the concentration of charge carriers can be controlled to adjust the refractive index and achieve certain colors. The color modulation in the designed structure was verified using the rigorous coupled wave analysis (RCWA) method. The angular independence of the relative transmission could reach up to 60°, and polarization-insensitive performance could also be maintained. Numerical results demonstrated that the thickness of IGZO was the key parameter to concentrate the incident light. The overall structure is planar and lithography-free and can be produced with simple preparation steps. The obtained results can also be extended to other similar resonators where a proper cavity allows dynamical functionality.

## 1. Introduction

As an important part of optoelectronic devices, color filters are widely used in color printing [1,2,3], high-resolution displays [4,5], sustainable color decoration [6,7] and safety colors [8]. Potential applications have aroused great interest in recent years. It is well-known that the color characteristics of traditional dye filters and pigment filters are determined by the inherent absorbance of their constituent materials [9,10]. In this context, the performance can be easily affected by high-intensity permanent light illumination and various chemical processes. As a result, the performance can be greatly reduced [11]. Color filters made from functional nanostructures rely on the interactions between light and the device, with the advantages of high efficiency, high spatial resolution and good stability [12].

Research on color filters has mainly been concentrated on areas such as surface plasmon resonance (SPR) [13,14], guided mode resonance (GMR) [15,16] and Mie resonance [17,18,19]. Transmitted or reflected color filters with dynamic management have been realized by changing the shape and structural parameters [20,21]. Although these nanostructure-based filters can achieve high performance, the color usually cannot be switched. The performance of nanostructures and thin films can be adjusted using the polarization state of incidence [22,23,24,25]. Phase change materials (PCMs) have also been widely used to achieve color switching. Ge_2_Sb_2_Te_5_ [26] and VO_2_ [27] are common PCMs, and their permittivity can be changed by controlling the temperature to produce a phase change. However, this effect occurs in the near-infrared band and cannot be freely applied to the visible light range. Furthermore, WO_3_ [28] and indium tin oxide (ITO) [29] can also undergo phase changes with changes in the applied voltage. The refractive index can also be influenced by the carrier concentration, which is much faster and more efficient [30].

In this paper, a new Fabry–Perot (F-P)-type transmissive color filter is proposed that can achieve structural colors with high efficiency using IGZO embedded between the metal Ag mirrors. By changing the thickness of the IGZO, high-purity structural colors can be obtained, and the carrier concentration can be changed using H_2_ treatment to control the refractive index of the IGZO, thereby achieving color switching in the visible range. The proposed color filter also presents incident angle-insensitive and polarization-independent performances. There are two advantages of this filter: high purity and faster switching speed.

## 2. Model and Methods

IGZO is a semiconductor that is widely used in neuromorphic electronic devices and flexible displays [31,32,33]. It has excellent electro-optical properties and many advantages with great potential for application in photonic devices. The electron density can be controlled by the H_2_ plasma treatment process, which generates free electrons as the injected hydrogen atoms form O-H bonds with the ionized oxygen in IGZO, as follows [34]:(1)$$H^{+}+O^{2-}={\mathrm{OH}}^{-}+e^{-}$$

The dielectric constant of IGZO is expressed by the Tauc–Lorentz–Drude model [35,36]:(2)$$\varepsilon_{\mathrm{IGZO}}=\varepsilon_{\mathrm{TL}}+\varepsilon_{D}=\varepsilon_{1}+i\varepsilon_{2}$$

Here, $\varepsilon_{\mathrm{TL}}$ and $\varepsilon_{D}$ represent the dielectric constants given by the Tauc–Lorentz and Drude models, respectively, and the Drude term describes the free electron absorption.

The complex permittivity of the Drude model is expressed as [34,36]:(3)$$\varepsilon_{D}(E)=-\frac{A_{D}}{E^{2}+\Gamma_{D}^{2}}-(\frac{A_{D}\Gamma_{D}}{E^{3}+\Gamma_{D}^{2}E})i$$

Here, *A_D_* is the amplitude and Γ*_D_* is the broadening parameter, respectively expressed as:(4)$$A_{D}=\varepsilon_{\infty }E_{p}^{2}=\varepsilon_{\infty }\hbar^{2}\omega_{p}{}^{2}$$
(5)$$\Gamma_{D}=\hbar \gamma ,\gamma =\frac{e}{m\;*\;\mu }$$

Here, *γ* represents the angular frequency, *μ* is the broadening parameter of light mobility, *E_p_* is the plasma energy, *ε*_∞_ is the high-frequency dielectric constant, *ℏ* is the Planck constant and *e* is the electronic charge. *ω_p_* is the plasma angular frequency, expressed as:(6)$$\omega_{p}=(\frac{e^{2}N}{m\;*\;\varepsilon_{\infty }\varepsilon_{0}}{)}^{1/2}$$
where *N* is the photocarrier concentration and *ε*_0_ is the free space permittivity.

The refractive index of IGZO can be changed through H_2_ plasma treatment. After treatment, the refractive index of IGZO decreases by ∼0.4, and the extinction coefficient also changes. This is due to the change in the electron density in the film. The electron density of the low-conductivity film is less than 10^14^ cm^−3^, while it is 8 × 10^19^ cm^−3^ after H_2_ is introduced, thus becoming a high-conductivity film [37]. Figure 1 shows the relationship between the dielectric constant and the incident wavelength of IGZO before and after the introduction of H_2_. The increase in electron density leads to a decrease in the dielectric constant.

The designed color-changing filter which involves inserting IGZO between the two metal layers, is shown in Figure 2a. Ag was selected as the metal reflector, as it has high reflectivity in the visible light range. Its refractive index was obtained from the literature [38]. For the scale parameters, the thickness of the top and bottom Ag was *t* = 30 nm, the thickness of the middle IGZO was *d* and the substrate was SiC. In the absence of free carriers, the optical properties of SiC are given by the Drude–Lorentz model [39]:(7)$$\varepsilon_{\mathrm{SiC}}=\varepsilon_{\infty }\frac{\omega^{2}-\omega_{L}^{2}+i\gamma \omega }{\omega^{2}-\omega_{T}^{2}+i\gamma \omega }$$

Here, $\omega_{L}$ and $\omega_{T}$ were selected as 972 cm^−1^ and 796 cm^−1^, respectively; $\varepsilon_{\infty }$ is the high-frequency dielectric constant, selected as 3.75 cm^−1^; and *γ* is defined as the damping rate caused by vibration anharmonicity, here set to 6.5.

Figure 3 shows the schematic of the multi-layer optical dielectric thin film with uniform dielectrics in each layer. We consider the reflection and the transmission of a TE-polarized plane wave (electric field perpendicular to the incident plane) of free space with wavelength *λ*_0_ and incident at angle *θ* on *L* uniform layers. The thin film above the high reflective films lies in the x-z plane. For analysis, the multi-layer dielectric can be divided into *L* layers along the *z* direction. Each layer possesses a refractive index of *n*_1_, *n*_2_ and *n_L_* and a thickness of *d*_1_, *d*_2_ and *d_L_*. The normalized electric field for the input and output regions can be written as:(8)$$\begin{array}{l} E_{I,y}=\exp\left[-jk_{0}n_{I}\left(\sin\theta x+\cos\theta z\right)\right]+\sum_{i}R_{i}\exp\left[-j\left(k_{x}x-k_{I,z}z\right)\right]\begin{array}{cc} & \left(Z<0\right) \end{array}\\ E_{II,y}=\sum_{i}T_{i}\exp\left\{-j\left[\left(k_{xi}x-k_{II,zi}\left(z-D\right)\right)\right]\right\}\begin{array}{cc} & \left(Z>D\right) \end{array} \end{array}$$
where *R* and *T* are the reflected and the transmitted amplitudes of the electric fields and *k*_0_ = 2π/*λ*_0_ is the wave-vector magnitude in the air. The wave vectors along the *x* and *z* directions in each divided layer are *k_xi_* = *k*_0_*n_i_* sin *θ* and *k_zi_* = *k*_0_*n_i_* cos *θ*, respectively.

The tangential magnetic and electric fields in the *m*-th (0 < *Z* < *D*) divided layer can be expressed in the following form:(9)$$\begin{array}{l} E_{m,y}=\sum_{i}S_{m,yi}\left(z\right)\exp\left(-jk_{xi}x\right)\\ H_{m,x}=-j{\left(\varepsilon_{0}/\mu_{0}\right)}^{1/2}\sum_{i}U_{m,xi}\left(z\right)\exp\left(-jk_{xi}x\right) \end{array}$$

Here, *ε*_0_ and *µ*_0_ are the permittivity and permeability of free space and *S_m_*_,*yi*_(*z*) and *U_m_*_,*xi*_(*z*) are the normalized amplitudes of the *m*-th space-harmonic fields that satisfy Maxwell’s equation in each divided layer.

As in the uniform, homogenous layer, the reflected and transmitted amplitudes can be solved by matching the tangential electromagnetic fields at the boundaries between each divided layer. At the boundary between the input region and the first layer (*Z* = 0), the following equation should be satisfied:(10)$$\left[\begin{array}{l} \delta_{i0}\\ jn_{1}\cos\theta \delta_{i0} \end{array}\right]+\left[\begin{array}{l} I\\ -jY_{1} \end{array}\right]\left[R\right]=\left[\begin{array}{cc} \begin{array}{l} W_{1}\\ V_{1} \end{array} & \begin{array}{l} W_{1}X_{1}\\ -V_{1}X_{1} \end{array} \end{array}\right]\left[\begin{array}{l} c_{1}^{+}\\ c_{1}^{-} \end{array}\right]$$

At the boundary between the *m*−1 and the *m* divided layer (*Z* = *D_m_*):(11)$$\left[\begin{array}{cc} \begin{array}{l} W_{m-1}X_{m-1}\\ V_{m-1}X_{m-1} \end{array} & \begin{array}{l} W_{m-1}\\ -V_{m-1} \end{array} \end{array}\right]\left[\begin{array}{l} c_{m-1}^{+}\\ c_{m-1}^{-} \end{array}\right]=\left[\begin{array}{cc} \begin{array}{l} W_{m}\\ V_{m} \end{array} & \begin{array}{l} W_{m}X_{m}\\ -V_{m}X_{m} \end{array} \end{array}\right]\left[\begin{array}{l} c_{m}^{+}\\ c_{m}^{-} \end{array}\right]$$

At the boundary between the last divided layer and the substrate (*Z* = *D_L_*):(12)$$\left[\begin{array}{cc} \begin{array}{l} W_{L}X_{L}\\ V_{L}X_{L} \end{array} & \begin{array}{l} W_{L}\\ -V_{L} \end{array} \end{array}\right]\left[\begin{array}{l} c_{L}^{+}\\ c_{L}^{-} \end{array}\right]=\left[\begin{array}{l} I\\ jY_{2} \end{array}\right]\left[T\right]$$
where W and V are a matrix whose element is determined according to the eigenvector and eigenvalues derived from Equations (1) and (2); Y_1_ and Y_2_ are diagonal matrices, with the diagonal elements being *k_Ι,zi_*/*k*_0_ and *k_ΙI,zi_*/*k*_0_, respectively, in each divided layer; and *c_L_* is the unknown constant to be determined. Therefore, the relation can be obtained as:(13)$$\begin{array}{l} \left[\begin{array}{l} f_{L}\\ g_{L} \end{array}\right]T_{L}=\left[\begin{array}{cc} \begin{array}{l} 1\\ \gamma_{L} \end{array} & \begin{array}{l} \exp\left(-k_{0}\gamma_{L}d_{L}\right)\\ -\gamma_{L}\exp\left(-k_{0}\gamma_{L}d_{L}\right) \end{array} \end{array}\right]\times \left[\begin{array}{l} a_{L}\\ b_{L}\exp\left(-k_{0}\gamma_{L}d_{L}\right) \end{array}\right]T_{L}\\ =\left[\begin{array}{l} a_{L}+b_{L}\exp\left(-k_{0}\gamma_{L}d_{L}\right)\\ \gamma_{L}\left[a_{L}-b_{L}\exp\left(-k_{0}\gamma_{L}d_{L}\right)\right] \end{array}\right]T_{L} \end{array}$$
where $\left[\begin{array}{l} a_{L}\\ b_{L} \end{array}\right]={\left[\begin{array}{cc} \begin{array}{l} 1\\ \gamma_{L} \end{array} & \begin{array}{l} 1\\ -\gamma_{L} \end{array} \end{array}\right]}^{-1}\left[\begin{array}{l} f_{L+1}\\ f_{L+1} \end{array}\right]$, and $f_{L+1}=1,g_{L+1}=jk_{II,z}/k_{0}$.

We can easily obtain the relation between reflected and transmitted amplitudes from Equations (4)–(7) without any numerical instability by using the enhanced transmittance matrix approach [6], as in the following:(14)$$\left[\begin{array}{l} \delta_{i0}\\ jn_{1}\cos\theta \delta_{i0} \end{array}\right]+\left[\begin{array}{l} I\\ -jY_{1} \end{array}\right]\left[R\right]=\left[\begin{array}{l} f_{1}\\ g_{1} \end{array}\right]T_{1}$$

Here, *f*_1_ and *g*_1_ are the assistant parameters in the enhanced transmittance matrix approach. From Equations (6) and (7), we can obtain the reflected amplitudes R and transmittance amplitudes T. Thus, the reflectance and transmittance of the multi-layer can be solved as follows:(15)R=RR*Re(kI,z/k0nIcosθ)T=TT*Re(kII,z/k0nIcosθ)

The entire structure is deposited on SiC, and a high-purity structural color can be obtained by changing the thickness of the IGZO. Figure 2c shows the transmission spectra corresponding to different IGZO thicknesses. When *d* = 170 nm, a blue (B) structural color appears at the resonance wavelength of 438 nm, and the transmission efficiency is 60.2%. When the thickness *d* of IGZO is adjusted to 90 nm and 125 nm, green (G) and red (R) structural colors with 55.9% and 49.9% transmission efficiencies at 550 nm and 676 nm can be obtained, respectively. Figure 2d plots the RGB chromaticity coordinates corresponding to the transmission curves in Figure 2c. The chromaticity coordinates for blue, green and red colors are (0.177, 0.072), (0.321, 0.502) and (0.450, 0.356), respectively. A wide-color RGB gamut is produced; thus, a variety of transmission colors can be generated.

## 3. Results and Discussions

The color filter exhibits the shape of a common F-P resonator, which is determined by the constructive interference and the destructive interference caused by the specific phase shift at the metal interface. Figure 4a,b depict the transmission spectra when *d* changes in the range of 80–180 nm before and after H_2_ treatment. The resonance wavelength shifts significantly with *d*. This wavelength is shifted after H_2_ treatment. The chromaticity diagram of the corresponding transmission curve is plotted in Figure 4c, and the chromaticity coordinates are shown in black and red, respectively. To visually demonstrate the performance of the color filter resulting from the change in IGZO electron density, we converted the transmission spectrum in Figure 4a,b into color, and the result is shown in Figure 4d. When *d* = 120 nm, the resonance wavelengths before and after the H_2_ treatment are 657.7 nm and 537.3 nm, respectively. The chromaticity coordinates change from (0.493, 0.326) to (0.284, 0.454), and the transmission color changes from red to green. For other values, the resonance wavelength will also shift after H_2_ treatment, and the transmission color will change. Therefore, by changing the electron density of the IGZO layer, the refractive index of the film can be changed so as to achieve the transformation of structural color.

During the design, the optimal design geometries for the MSM cavity had to be obtained. To achieve this goal, the transmission spectra for different IGZO thicknesses before and after H_2_ treatment at normal incidence were determined. The structures were excited with a plane wave in the frequency range of interest, which was 400–800 nm. As shown in Figure 5a,b, the resonant peak was obtained by gradually increasing the thickness of IGZO under TM polarization in the visible range considered. The shift in the resonance wavelength before and after H_2_ treatment showed a tendency to shrink, and the transmission effect was weakened. High-purity RGB colors were obtained, as shown in Figure 5a, and the resonance peak could be shifted by adjusting the thickness. Therefore, the resonance wavelength shifts significantly after H_2_ treatment. For TE-polarized light, the same result was obtained under TM polarization, as shown in Figure 5c,d. The polarization of the implemented filter was independent.

The thickness of the metal layer reflector affects the transmission effect and the color produced. In the following, we study the effect of the metal layer thickness *t* on the transmission spectrum and color-rendering properties. Figure 6a depicts the change in the transmittance of RGB with the thickness of Ag at normal incidence when the filter presents the G color. As *t* increases, the transmission efficiency decreases and the resonance wavelength is red-shifted. Figure 6b shows the transmission spectrum after H_2_ treatment; with the increase in *t*, the transmission efficiency also decreases, but the spectral response additionally becomes clearer, resulting in improved color purity. As shown in Figure 6c, this is the color coordinate described in the CIE 1931 chromaticity diagram. When *t* = 20 nm, although the transmission efficiency is very high, it deviates from the original green. The H_2_ treatment also has the same result. The variation in the resonance results in a deviation from the original blue color. Therefore, we must fully consider the influence of the thickness of the reflector on the color-rendering effect.

In order to study the physical properties of the adjustable structure before and after H_2_ treatment, the electric field distribution diagram at the resonance wavelength of 550 nm (G color) was studied. Figure 7a,b show the electric field changes before and after H_2_ treatment. It can be seen that the highest electric field intensity in the low-conductivity film reaches 2.7, which is significantly stronger than that of the high-conductivity film. The electric field in Figure 7a is mainly concentrated at the junction of the metal cavity and IGZO and FP resonance occurs, enhancing transmission efficiency, while the light field in Figure 7b is mainly located in the top Ag cavity, which leads to weakened resonance and a weakened transmission effect.

Next, we studied the relationship between the transmission spectra before and after the H_2_ treatment and the incident angle. From Figure 8a,c,e, it can be seen that, for low-conductivity films, within the incident angle range of 0–60°, the resonance wavelength remains almost unchanged, and the transmission efficiency does not change much. High-purity RGB structural colors can be obtained and good resonance characteristics maintained. When the B color appears, as the incident angle increases, resonance appears at 780–800 nm, but the color filter effect is not affected. Figure 8b,d,f correspond to the transmission spectra of IGZO treated with H_2_ shown in Figure 8a,c,e, respectively. The position of the resonance peak shows a significant color shift. When the incident angle changes from 0 to 60°, the resonant peak position of RGB color is almost unchanged. It is thus proved that the structure has good incident angle insensitivity.

## 4. Conclusions

In summary, numerical research was carried out on ultra-thin and efficient color-changing filters. The Ag-IGZO-Ag resonator covering the IGZO layer was used to achieve FP resonance. An adjustable bandwidth in the fixed-resonance wavelength could be realized, and any color in the visible light band could be obtained. The output color had the advantages of rich color, wide color gamut, high color saturation and high purity. Through the H_2_ treatment, the IGZO could become a high-conductivity film, so that the color could be switched, the resonance wavelength shifted and a transmission palette obtained. Moreover, it was determined from the RCWA that, when the incident angle was changed between 0 and 60°, the wavelength of the resonant peak remained basically unchanged; thus, it has good incident angle-insensitive characteristics. The proposed structure has significant application prospects and great potential for displays, image sensors and decorations.

## Figures and Tables

**Figure 1 micromachines-13-01228-f001:**
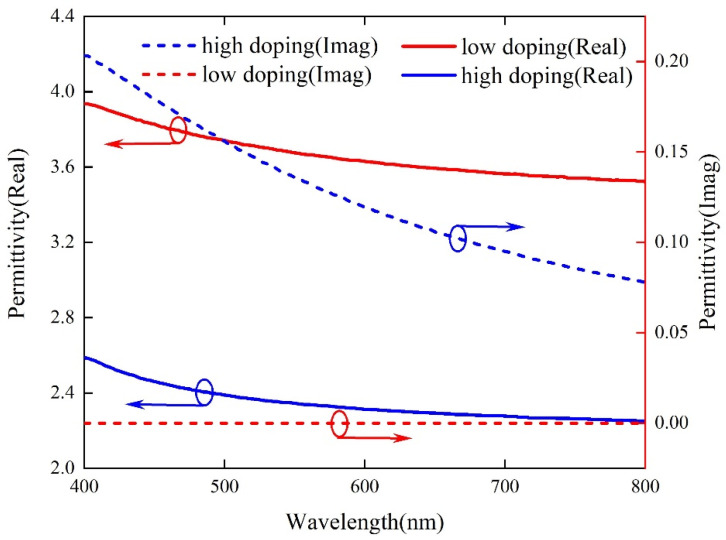
The real and imaginary parts of the complex dielectric constant of IGZO vary with wavelength.

**Figure 2 micromachines-13-01228-f002:**
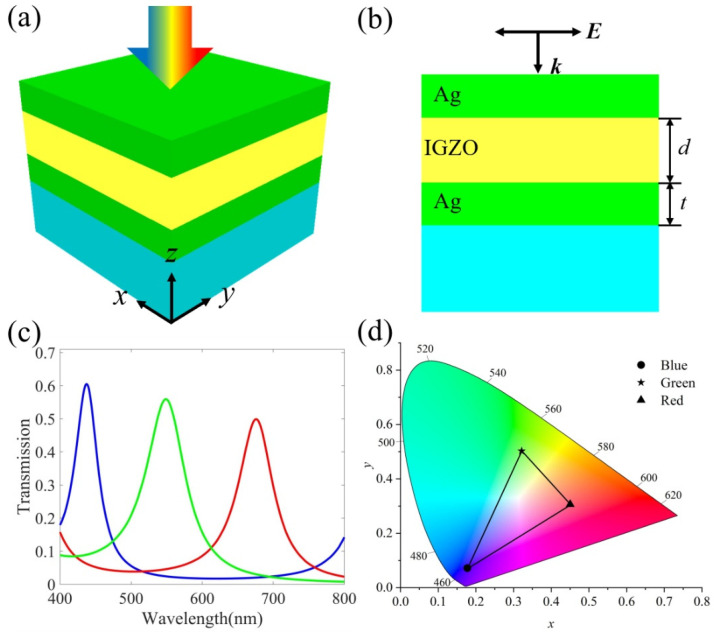
(**a**) Schematic diagram of the proposed ultra-thin, high-efficiency color-changing filter; (**b**) cross-sectional view of the structure; (**c**) the calculated transmission spectrum at normal incidence. *T* = 30 nm and *d* = 170 nm, 90 nm and 125 nm, respectively; (**d**) the corresponding chromaticity diagram of the RGB structural color.

**Figure 3 micromachines-13-01228-f003:**
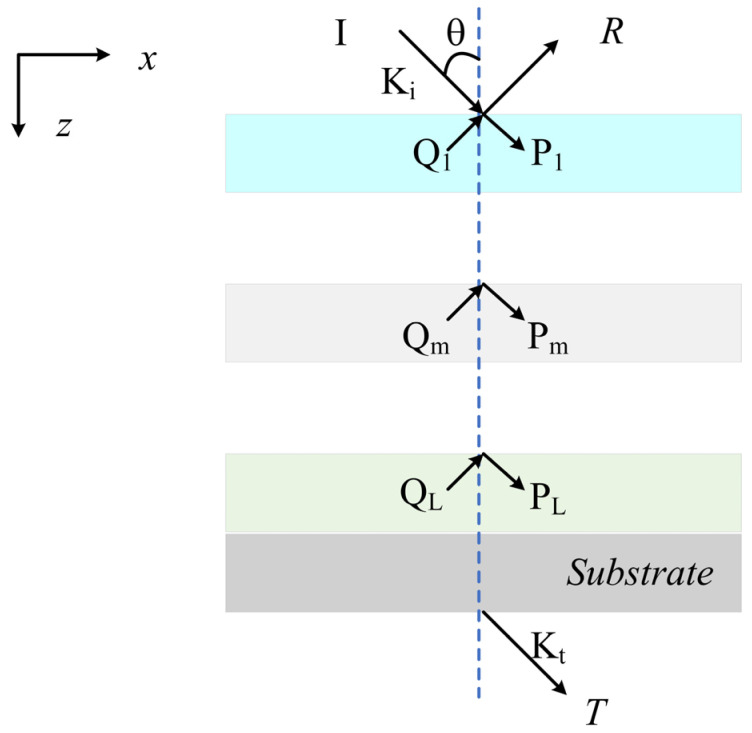
Geometry for the reflection and the transmission from a stack of multi-layer optical dielectric thin films.

**Figure 4 micromachines-13-01228-f004:**
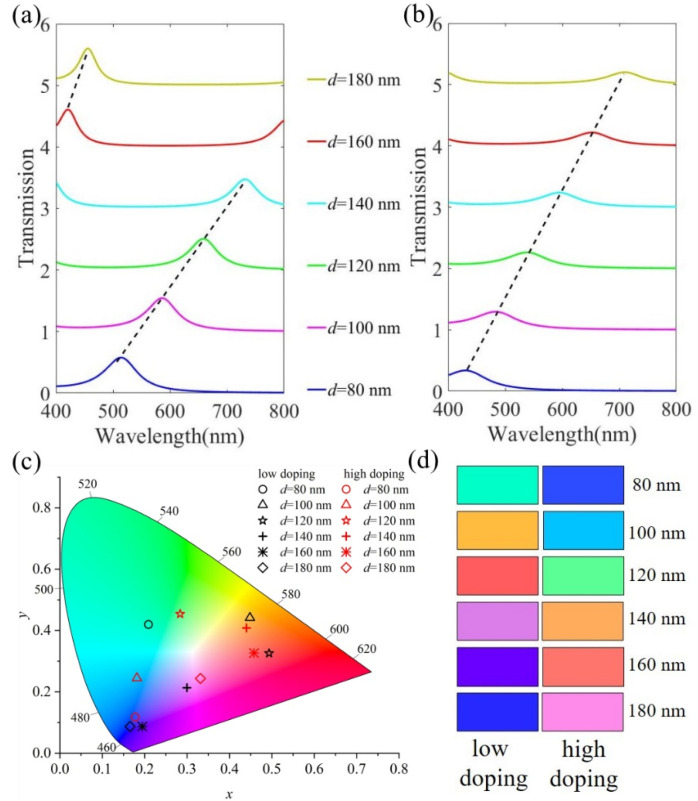
Simulated spectral transmittance curves at normal incidence with different *d* (**a**) before H_2_ plasma treatment and (**b**) after H_2_ plasma treatment. (**c**) The corresponding chromaticity diagram for RGB structural colors before and after H_2_ plasma treatment. (**d**) The color change corresponds to the (**a**,**b**) transmission curve.

**Figure 5 micromachines-13-01228-f005:**
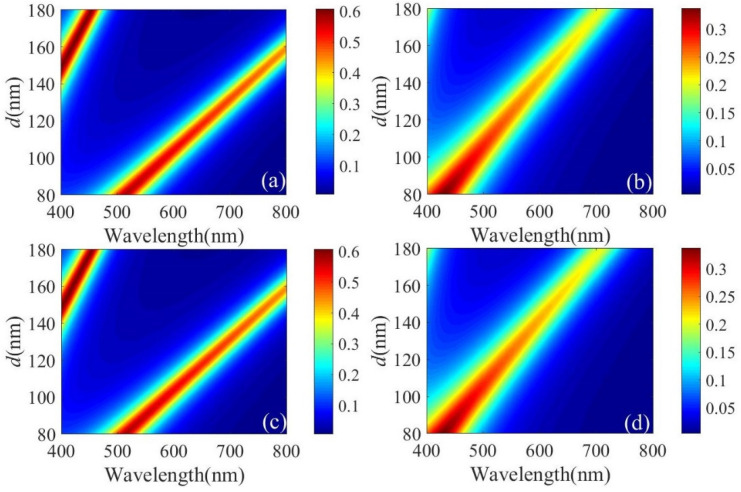
For TM polarization: (**a**) transmission spectra of different IGZO thicknesses before H_2_ treatment and (**b**) after H_2_ treatment. For TE-polarization: (**c**) transmission spectra of different IGZO thicknesses before H_2_ treatment and (**d**) after H_2_ treatment.

**Figure 6 micromachines-13-01228-f006:**
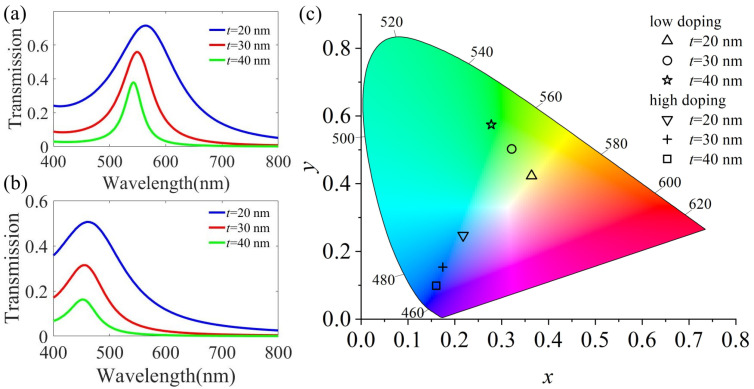
(**a**) Simulated spectral transmittance curves at normal incidence with different *t* for green, *d* = 90 nm. (**b**) The corresponding transmission spectrum after plasma treatment with H_2_. (**c**) An illustration of color coordinates calculated from the transmission spectra studied in (**a**,**b**).

**Figure 7 micromachines-13-01228-f007:**
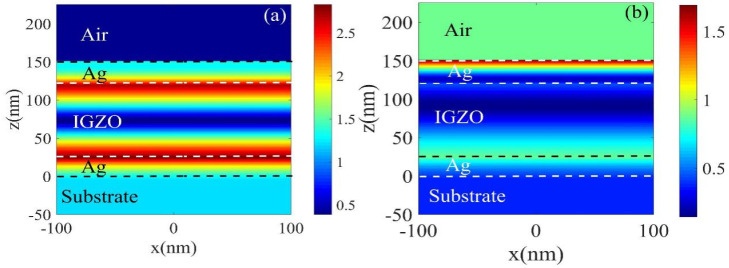
The electric field distribution curve for the color filter (green) with a resonance wavelength of 550 nm in the x−z plane, *t* = 30 nm, *d* = 90 nm: (**a**) before H_2_ treatment; (**b**) after H_2_ treatment.

**Figure 8 micromachines-13-01228-f008:**
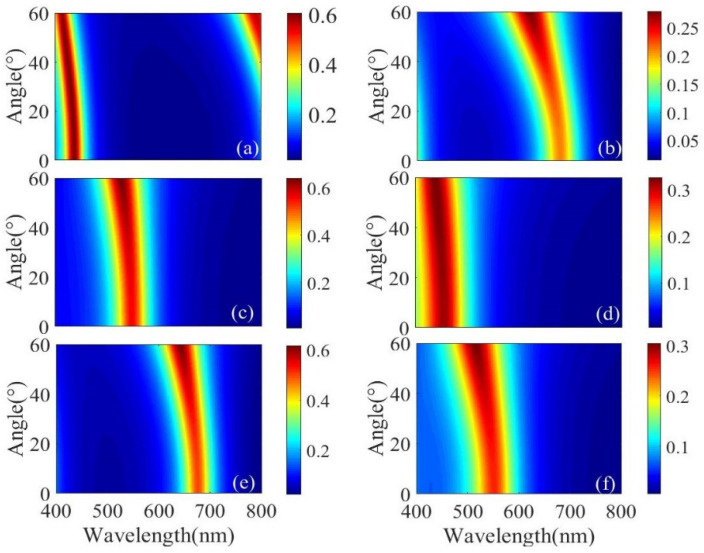
For low-conductivity films, the transmission spectra are functions of the wavelength and incident angle of (**a**) B, (**c**) G and (**e**) R colors, respectively. For high-conductivity films, the transmission spectra are functions of the wavelength and incident angle of (**b**) R, (**d**) B and (**f**) G colors, respectively.

## Data Availability

The data that support the findings of this study are available from the corresponding author upon reasonable request.

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
