# Peer review of "Ultra-Thin and Lithography-Free Transmissive Color Filter Based on Doped Indium Gallium Zinc Oxide with High Performance"

_micromachines, 2022, doi:10.3390/mi13081228_

Round 1
Reviewer 1 Report
1. As described in the model and methods’ section, the refractive index of IGZO could be changed by H2 plasma treatment. It is also mentioned below that the refractive index decreased by 0.4 after H2 treatment. However, it does not indicate whether a 0.4 drop of refractive index at a particular wavelength or an average value within the particular band. Therefore, it is suggested to supplement the curve of refractive index and wavelength (before and after H2 treating).
2. When d=170nm, the blue structural color will be exhibited. However, as the Fig2(a) shown, the line is blue when d=80nm, the blue line is represented for the structural blue or not. It is suggested that the color of the curve corresponds to the corresponding structural color.
3. In Fig.5(a), (b), the value of t and d should be indicated.
4. The simulation research and discussion of structural color are based on RCWA model, however, the simulation software and simulation conditions are not described. Meanwhile, is the simulation of electromagnetic field also based on RCWA model?
5. Is there a real rendering of the sample at different wavelengths?
6. All pictures are blurred.
Author Response
We have carefully read the reviewer’s reports on our manuscript and would like to express our sincere thanks to the reviewer for bringing us the helpful comments.

Reviewer 2 Report
Manuscript from Xiangrui Fan et al. deals with transmissive color filter based on doped indium gallium zinc oxide. My overall feeling is that the quality of the manuscript is low. The main idea of the manuscript (e.g. changing of wavelength corresponding to the maximum of the transmittance by the film thickness) is the topic of the physics classes in the first year of Bachelor studies.
Negative aspects that leads me to the recommendation to reject the manuscript:
1. Only theoretical study is presented. If supported by at least one sample / layer which have predicted transmission it would improve the manuscript quality / relevance / reliability significantly.
2. Why SiC is chosen for substrate? Do you personally use color filter on SiC? For visible part of spectrum maybe glass would be better?
3. In appropriate software (e.g. WASE 32) with known optical properties you can model transmission curves shown in fig. 1 c within 5 minutes. Moreover you can try different layers (e.g. ZnO instead of IGZO) to see a possible difference.
4. Which software was used for rigorous coupled wave analysis (RCWA)? Does finite element method (e.g. COMSOL) give you the same results?
To conclude, based only on theoretical approach I am able to model data presented in manucript in 10 minutes. Without at least experimentally deposited / prepared sample I do not consider the quality of the manuscript to be sufficient for publication.
Author Response

(The authors gave the same response as above.)

Reviewer 3 Report
In this manuscript, a new Fabry-Perot type transmissive color filter using the structure of Ag/IGAO/Ag is proposed. In my opinion, this manuscript is interesting to the readers of Micromachines. The topic is very important in this field. This work is novel and original. The authors have solid background in this field. Therefore, the referee recommends it to be published after the following minor revisions:
1. The English should be polished by a native speaker.
2. The background (review) about IGZO and its applications should be provided in the introduction for the readers in this field. Please cite more recent reports. For example, ACS Applied Electronic Materials, 2, 2976-2983, 2020; ECS Journal of Solid State Science and Technology, 11, 067001, 2022.
3. Fig. 2 and Fig. 4. Please provide this graph with better quality.
4. As well as by band-to-band excitation, electrons are also generated as a result of deep-level-to-band excitation. The presence of deep-level traps in the middle of the bandgap (~2.25 eV) has already been verified by previous studies.
Refer to:Curr. Appl. Phys. 11 280, 2011 & ACS Photonics 2 1057, 2015.
Does the presence of deep-level traps affect the results in this study?
In general, this work seems to be very interesting. The referee would like to see the revision if possible.

Author Response

(The authors gave the same response as above.)

Round 2
Reviewer 1 Report
The authors addressed my comments and improved their manuscript. I recommend it for publication.
Author Response
Thanks for the recommendation of our paper.
Reviewer 2 Report
I appreciate that authors would consider my comment about only theoretical approach. I am convinced that experimental data would increase the quality / relevance / scientific soudness of the article. Without experiment I do not consider quality of the manuscript to be good enough to be published.
In the response authors replied: "The simulation in this paper is carried out using homemade matlab codes based on rigorous coupled wave analysis"
My question is why the theoretical backround on this is missing? Why you do not insert matlab codes and also the theory into the manuscript?
I have no idea how you construct your matlab codes.
Moreover if "comparisons have been made with FEM and FDTD methods." why this comparison is missing in the manuscript.
To conclude, I reccomend authors to carefully rewrite the manuscript, conclude also experimental measurement and after this I beleive that the manuscript can be easily published.
Author Response
We have carefully read the reviewer’s reports on our manuscript and would like to express our sincere thanks to reviewer #2 for bringing us the helpful comments. We have accepted all the appropriate suggestions to modify our manuscript. We would like to briefly describe the changes that we have made and present some explanations to response the reviewer’s comments.

Reviewer 3 Report
The authors have properly addressed my questions. Thus, I recommend its publication in its present form.
Author Response

(The authors gave the same response as above.)

Round 3
Reviewer 2 Report
Although if I consider the manuscript as a theoretical work I have strong negative points against its publishing:
1. Idea of Ultra-thin and lithography-free transmissive color filter can be found in the literature.
Trying to find similar topics I found several articles e.g.X. Zheng et. al. High-color-purity transmissive colors with high angular tolerance based on metal/dielectric stacks. Optics Communications 434 (2019) 70–74 describing similar idea as authors with Ag + ZnS + Ag structure or conference paper Theoretical Study of a Tunable Transmissive Subtractive Color Filter UsingThin Metal films and Transparent Conducting Oxide from K. Jungsan et al. (DOI: 10.1109/OECC.2018.8729898) or work of J. Lin et al.: Multilayer structure for highly transmissive angle-tolerant color filter. Optics Communications 427 (2018) 158–162.
Therefore authors should clearly define novelty of their manuscript. From my point of view the novelty is not well claimed.
2. Even more negative is that the same authors published several works on this topic. Especially
S. Wang et al. High saturation structural color based on lithography-free planar thin film with doped indium-gallium-zinc-oxide semiconductor. Optik - International Journal for Light and Electron Optics 255 (2022) 168731 (https://www.sciencedirect.com/science/article/pii/S003040262200136X?via%3Dihub).
There is high percentage of accordance between the manuscript submited to Micromachines and this article.
E.g. Figure 1 in the manuscript is the same as figure 1 in above mention article and you will have the structure in the manuscript considering thickness of Ge layer (h in the above mentioned manuscript) as zero.
I am really sorry but I cannot reccomend your manuscript for publication as I am convinced that it is kind of self-plagiarism. Your manuscript does not bring any new information in comparison with your work published in Optik and you even do not mention this article in your manuscript.